# Hospitalization Costs and Financial Burden on Families with Children with Depression: A Cross-Section Study in Shandong Province, China

**DOI:** 10.3390/ijerph16193526

**Published:** 2019-09-20

**Authors:** Yawei Guo, Jingjie Sun, Simeng Hu, Stephen Nicholas, Jian Wang

**Affiliations:** 1Center for Health Economics Experiment and Public Policy, School of Public Health, Shandong University; Key Laboratory of Health Economics and Policy Research, NHFPC (Shandong University), Jinan 250012, China; sddxgyw@163.com (Y.G.); SimengHu124@163.com (S.H.); 2Shandong Health Commission Medical Management Service Center, Jian 250014, China; sunjingjie163@163.com; 3School of Management and School of Economics, Tianjin Normal University, West Bin Shui Avenue, Tianjin 300074, China; 4Newcastle Business School, University of Newcastle, University Drive, Newcastle, NSW 2038, Australia; 5Guangdong Institute for International Strategies, Guangdong University of Foreign Studies, Baiyun, Guangzhou, Guangdong 510420, China; 6Top Education Institute, 1 Central Avenue, Australian Technology Park, Eveleigh, Sydney, NSW 2015, Australia; 7Dong Fureng Institute of Economic and Social Development, Wuhan University, No.54 Dongsi Lishi Hutong, Dongcheng District, Beijing 100010, China; 8Center for Health Economics and Management at School of Economics and Management, Wuhan University, 299 Bayi Road, Wuchang District, Wuhan, Hubei Province 430072, China

**Keywords:** hospitalization costs, financial burden, childhood depression, China

## Abstract

Background: Depression, one of the most frequent mental disorders, affects more than 350 million people of all ages worldwide, with China facing an increased prevalence of depression. Childhood depression is on the rise; globally, and in China. This study estimates the hospitalization costs and the financial burden on families with children suffering from depression and recommends strategies both to improve the health care of children with depression and to reduce their families’ financial burden. Methods: The data were obtained from the hospitalization information system of 297 general hospitals in six regions of Shandong Province, China. We identified 488 children with depression. The information on demographics, comorbidities, medical insurance, hospitalization costs and insurance reimbursements were extracted from the hospital’s information systems. Descriptive statistics were presented, and regression analyses were conducted to explore the factors associated with hospitalization costs. STATA14 software was used for analysis. Results: The mean age of children with depression was 13.46 ± 0.13 years old. The availability of medical insurance directly affected the hospitalization costs of children with depression. The children with medical insurance had average total hospitalization expenses of RMB14528.05RMB (US$2111.91) and length of stay in hospital of 38.87 days compared with the children without medical insurance of hospital with expenses of RMB10825.55 (US$1573.69) and hospital stays of 26.54 days. Insured children’s mean out-of-pocket expenses (6517.38RMB) was lower than the those of uninsured children (RMB10825.55 or US$1573.69), significant at 0.01 level. Insured children incurred higher treatment costs, drug costs, bed fees, check-up fees, test costs and nursing fees than uninsured patients (*p* < 0.01). Conclusions: Children suffering from depression with medical insurance had higher hospitalization costs and longer hospitalization stays than children without medical insurance. While uninsured inpatients experienced larger out-of-pocket costs than insured patients, out-of-pocket hospital expenses strained all family budgets, pushing many, especially low-income, families into poverty—insured or uninsured. The different hospital cost structures for drugs, treatment, bed fees, nursing and other costs, between insured and uninsured children with depression, suggest the need for further investigations of treatment regimes, including over-demand by parents for treatment of their children, over-supply of treatment by medical staff and under-treatment of uninsured patients. We recommend more careful attention paid to diagnosing depression in girls and further reform to China’s health insurance schemes—especially to allow migrant families to gain basic medical insurance.

## 1. Introduction 

Mental disorders are a serious illness that can occur at any age in a person’s life, and compared with other medical diagnoses, mental disorders are very common [1]. As one of the most frequent mental disorders, depression affects more than 350 million people of all ages worldwide, with, on average, one in 20 people reporting an episode of depression in the previous year [2,3,4]. The World Health Organization (WHO) ranked depression the fourth leading cause of disability worldwide and projected that by 2020, depression will climb to be the second leading disability [5,6]. The WHO estimated as well that 10% to 20% of children experience mental disorders globally [7]. Given its high prevalence, long duration, early onset and serious complications, mental disorders place a particularly serious burden on children [8]. About half of all lifetime mental diseases start during adolescence, which are accompanied by affective disorders with the highest lifetime prevalence, including depression [9,10]. Mental disorder incidents occurring in early stages of a person’s whole lifetime have serious and deleterious long-term sequelae [11]. An Australian study found that mental disorders were the “largest” contributor to disabilities in adolescents, followed by chronic respiratory diseases and neurological disorders [12] and mental disorders underlie a substantial proportion of suicides—a leading cause of death in youths [13].

China faces the challenge of the increasing prevalence of depression as well, which has become one of the leading causes of disability-adjusted life years in China [14,15,16]. Not only has the prevalence of depression been increasing over the past decades [15], but more and more people are suffering from depression in childhood [4]. Childhood is an important period in an individual’s life when he/she establishes personal identity and completes education. Without effective treatment, depression can severely impact children’s development, their educational attainment and their potential to live fulfilling and productive lives. We know depression in children is associated with long-term mental and physical health problems, and, if untreated, depression increases the risk of self-harm and suicide. For example, approximately two-thirds of suicide completers or attempters have experienced major depressive episodes at the time of the suicidal act [17,18,19,20].

Besides a serious medical condition, depression can impose a heavy financial burden both on families and the society, including significant resource demands on a country’s health system [2,21,22,23], there is a three-tiered public-private hospital system in China, comprising village clinics, township hospitals and county hospitals in rural areas, and community health centers, district hospitals and municipal hospitals in urban areas [24]. Village clinics, township hospitals and community health centers provide preventive and primary care services; 100-499 bed county and district hospitals provide secondary specialty care and inpatient services; and large-scale city-based tertiary hospitals, with over 500 beds, provide complex healthcare [25]. European scholars have pointed out that major depression was among the most expensive disorders in relation to the total costs of brain disorders in 2010 [26]. An American analysis of the financial burden of depression found the annual costs totaled US$53 billion [27]. A survey of 28 European countries estimated the cost of depression to be €118 billion (US$131.42 billion) in 2005, accounting for 13% of the total health care expenditure of these countries [28]. Other studies have shown that depression costs average 1000 to US$2500 per person per year across western developed countries [29], and a South Australian study estimated the direct medical cost per patient with depression as US$2454 [30]. Depression can significantly increase hospitalization costs for other illnesses as well. A study in Tennessee found that depression can lead to significantly higher hospitalization costs for patients with lung cancer, heart failure and diabetes [31,32]. Many poor patients with depression cannot get timely and effective treatment, which frequently leads to individual and socially serious consequences, such as suicide or anti-social behavior [33]. In one of the few Chinese studies, about 6.9% of total personal medical expenditure was attributable to depression and about 7.8% of total medical expenditure was attributable to the depressive symptoms [23], which is almost three times as large as the impact of obesity on health care costs [34].

While previous Chinese studies have demonstrated the significant financial burden of depression, the focus has largely been on adults [35,36,37,38], a large number of Chinese studies have explored depression costs related to concomitant diseases [15,39,40,41,42,43]. Few Chinese studies have explored the hospitalization costs and the financial burden on families of children with depression. This study estimates the hospitalization costs and the financial burden on families with children suffering from depression and recommends strategies both to improve the health care of children with depression and to reduce their families’ financial burden.

## 2. Materials and Method

### 2.1. Settings and Data Source

The study was jointly completed by the School of Public Health of Shandong University and the Health Committee of Shandong Province, China. A three-stage cluster sampling method was used to select the sampling sites. First, based on the regional economic development of Shandong Province, the province was divided into three regions: eastern, middle and western. As shown in Figure 1, two cities were selected in each region, with Qingdao and Weifang in the eastern region, Jinan and Linyi in the middle region and Dezhou and Jining in the western region. By drawing four districts or counties in each extracted city, the second stage sampling balanced urban and rural areas. Finally, eight communities and townships were selected in each of the districts and counties. The final sample comprised urban and rural patients across the three regions, including 192 communities or townships drawn from 24 districts or counties in six cities. From these 192 sampling sites, we selected 297 health facilities as sampling institutions, including general hospitals, traditional Chinese medicine hospitals, maternal and child health hospitals, specialist hospitals, community health service centers and township health centers. From the 297 health facilities, we extracted data on all inpatients from 1 January 2017 to 31 December 2017.

### 2.2. Study Population

Hospitalized patients with depression were identified using the International Classification of Diseases, Tenth Revision (ICD-10) diagnosis codes F32–F33. All diagnoses were made by the attending physicians. Using the electronic system of the medical facility, data were uniformly reported by trained medical staff. In order to reduce any errors, all ICD-10 codes were proofread using a special computer program developed by the National Health and Wellness Development Committee. Patients aged less than 18 years with at least one day of hospitalization were included, yielding a sample of 488 children with depression from 297 health facilities across Shandong province. All eligible patients had complete information regarding their demographics, comorbidities, sex, age, health insurance, hospitalization costs, insurance reimbursements and the information about the health facilities.

China is in the process of integrating the New Cooperative Medical Scheme (NCMS), for the rural population, and the Urban Residents Basic Medical Insurance System (URBMI), for the urban population out of the workforce, including urban children, students, the unemployed and the disabled, into the Integrated Urban and Rural Medical Insurance Scheme (IURMI). The Shandong province completed the reform in 2015. For children under the age of 18, they only had IURMI or no insurance.

### 2.3. Statistical Analysis

Statistical analysis was performed using STATA Version 14.0, with statistical significance set at the 5% significance level. For continuous variables, the p value was calculated using the Student’s t test; for categorical variables, *p* value was calculated using the chi-square test. We used a multivariate linear regression model to analyze relationships between hospitalization costs and potential influential factors. We specified the following regression model:Y = β0 + β1 × X1 + β2 × X2 + ···+ βm × Xm + e(1) where the dependent variable was hospitalization costs and the independent variables were age, sex, with or without medical insurance, with or without comorbidities, geographical location and type of the health facility.

## 3. Results

### 3.1. Basic Information on the Participants

As shown in Table 1, of the 488 children with depression under the age of 18 in Shandong Province, 76 were from the eastern, 173 from the middle and 239 from the western region. The male (252) to female (236) ratio was close to 1:1, with the youngest patient aged 9 years old, 145 patients aged 9–11 years, 151 patients aged 12–14 years and 192 patients aged 15–17 years. Roughly 47%, or 227 children, had various types of comorbidities. Three hundred and sixty-four (364) children had IUBMI, while 124 children were uninsured, paying their own medical expenses, and 438 were from national public hospitals and 50 were from private health facilities.

### 3.2. Factors Associated with Total Hospitalization Costs, Length of Stay and Out-of-Pocket Expenses

We measured total hospitalization costs, length of hospital stay (LOS) and out-of-pocket (OOP) expenses of children with depression by sex, age, region, comorbidities, insurance and type of health facility. As shown in Table 2, there were five groups with statistically significant differences (*p* < 0.05): the LOS of patients by region and medical insurance; OOP expenses by type of health facility and with/without insurance; and total hospitalization costs by with/without insurance.

In Table 3, the results of the multiple linear regression show that the availability of medical insurance had a significant impact on the total hospitalization cost (*p* < 0.01), with the total hospitalization costs of patients with medical insurance significantly higher than that of patients without insurance. Comorbidities, age, sex of patients, region of the health facility had no significant association with the total hospitalization costs.

### 3.3. Hospitalization Costs of Children with Depression with and without Medical Insurance

As shown in Table 4, insured patients suffering from depression had significantly higher total hospital expenditure (RMB14528.05 ± 490.26/US$2111.91 ± 71.27) than those without medical insurance (RMB10825.55 ± 674.39/US$1573.69 ± 98.03, *p* < 0.001) and had a longer LOS in hospital (38.87 ± 1.52 days) than those without medical insurance (26.54 ± 1.89 days, *p* < 0.001). As expected, patients suffering from depression with medical insurance had significantly lower OOP expenditure (RMB6517.38 ± 239.81) than those without medical insurance (RMB10825.55 ± 674.39/US$1573.69 ± 98.03, *p* < 0.001). Breaking down the specific hospital expenditures in Figure 2, treatment, drug, bed fees, medical checks, nursing, tests and other costs were all significantly higher for children with medical insurance than those without insurance. Only the diagnose expenditure were not significantly different between insured and uninsured inpatients.

## 4. Discussion

This is the first study to explore the composition of hospitalization expenses for children with depression with or without medical insurance. Our study found that the average hospitalization costs for children with depression was RMB13587.25 (US$1828.96). A systematic review of 24 papers on the cost-of-illness studies of depression in western developed countries found that the direct hospital cost per patient ranged between US$1000 and US$2500 [29]. An Australian study estimated that the average direct cost of hospitalization for depression was US$2454 [30]. Hospitalization costs of adult American company workers with depression was estimated to be US$1341 [44]. These non-Chinese studies of hospitalization costs are broadly similar to our estimates for Chinese children with depression. However, a U.S. study of average hospitalization cost of depression patients over the age of 65 was US$18967 [45], which was significantly higher than our estimate for Chinese children. The reason was most likely related to elderly patients having complicated additional physical health conditions, leading to a sharp increase in hospitalization costs.

Previous studies of the costs of depression have shown that the presence or absence of comorbidities significantly effect hospitalization costs [46]. However, our study found that there was no significant effect of the comorbidity on the medical expenses (*t* = 1.311). We speculate that the physical condition of the children is relatively better than that of the adults, with fewer comorbidities or more mild comorbidities, which significantly reduced hospitalization expenses and length of hospital stays for children.

According to the Shandong provincial commission of health and family planning, the proportion of men and women under the age of 18 is close to 1:1. Previous studies have shown that the prevalence of depression in women is twice that of men [1]. However, in our study, we found that there was no significant difference in the proportion of male and female children with depression (*p* < 0.01). There may be various explanations for the lack of a sex difference in children with depression. Shandong females may have suffered from less depression than the estimate in other studies. Alternatively, female depression may have been underdiagnosed or females with depression may have made fewer hospital visits. Given China’s traditional value system, which ‘values’ boys differently than girls, many families pay more attention to the health of boys than girls. In this scenario, some girls suffered from depression, but went undiagnosed. Another possibility is that girls were more introverted, leading to their depression to be found by their parents or others rather than self-identified. There is a need to more clearly identify depression in female children, which may require better training by medical professionals to identify depression in girls or to educate females and their families to be vigilant for signs of depression.

Our study found that the existence of medical insurance had a significant impact on the hospitalization expenses of children with depression (*t* = 3.923). Specific medical expenses in Table 3, such as treatment, drugs, bed fees, health check expenditure, nursing costs and test expenditures, were significantly higher for patients with medical insurance than for children without medical insurance (*t* < 0.05). Insured children with depression had significantly longer hospital stays as well than children without insurance (38.87 ± 1.52 days versus 26.54 ± 1.89 days). One conclusion is that children without medical insurance did not receive adequate treatment due to hospitalization expenses. Without effective treatment, depression can have serious consequences [47]. Alternatively, insured patients may have been over-serviced, which wasted limited medical resources. The level of doctors’ medical treatment and professional quality should be standardized to avoid the occurrence of moral hazard related to over-servicing. Additionally, there is a strong case to expand the coverage of basic medical insurance, ensuring that more children with depression get effective treatment and alleviate the disease burden of depression [48]. We speculate that most uninsured children came from migrant households ineligible for insurance or poor households unable to afford insurance. The hukou system of household registration links medical insurance and health care to the family’s rural address, denying migrant families—even with insurance—access to hospitals outside their original rural residence area.

Medical insurance reduced the financial burden of hospitalization costs for insured families, while uninsured inpatients faced significant hospitalization costs that severely strained average household budgets. From Table 4, depression patients with medical insurance had significantly lower OOP expenses (RMB6517.38 ± 239.81/US$947.42 ± 34.86) than patients without medical insurance (RMB10825.55 ± 674.39/US$1573.69 ± 98.03, *p* < 0.001). In 2015, the average annual income of middle-income households in China was RMB49809.73 (US$7239.05) and that of low-income households was only RMB12048.79 (US$1751.10). Average out-of-pocket expenses of RMB6517.38(US$947.65) for insured middle-income families accounted for about 13% of annual household income but accounted for 54% of annual household income of low-income families. For uninsured families, out-of-pocket expenses of RMB10825.55(US$1574.05) for middle-income families was 22% of the annual household income and 90% of the annual household income for low-income families. Both insured and uninsured low-income families’ financial resources would be severely impacted by out-of-pocket expenses. One measure of medical poverty is when over 40% of a family’s yearly non-food budget was accounted for by out-of-pocket hospital expenses [49]. Insured low-income families faced OOP expenses accounting for 54% of their total household income and uninsured families 90% of their total household income—plunging these families into poverty. Families in poverty were forced to sell assets or borrow funds from family and friends to pay off OOP medical bills. The Chinese government should strengthen the social security fund, expand the coverage of social security and lower the threshold fees for insurance, especially for migrant and poor families not covered by health insurance. Some mainly rural families, poorly educated and remote, may not understand the advantages of health insurance, so the government might mount education campaigns on the benefits of joining insurance schemes.

We undertook some broad estimates of the total cost of childhood depression on China’s health system as well. Chinese economic development is uneven, with the eastern coastal areas being the most developed, the central region displaying average levels of development, and the western region being relatively backward [50]. The economic development of the three regions in Shandong province is similar to that of China, with Shandong known as the epitome of China [51]. On this basis, our Shandong data can be extended to the whole China. Table 5 estimates the costs of hospitalization of children with depression accounted for 0.016% or RMB6630578 (US$961916) of China’s total health budget or about 7% of the total hospital costs for all patients with depression.

There are limitations to our study. First, it was a cross-section study, and the relationship between hospitalization costs and the associated factors cannot be interpreted as cause and effect. Second, there was a lack of information on the detailed symptoms of patients, so we were unable to analyze whether the hospital and medical cost were ‘reasonable’ for the treatment required. Further studies should collect more socioeconomic characteristics as well, such as household income and wealth, which will allow for a better estimation of the financial impact of children with depression on families. While the results from this study are representative for the Shandong province, further studies from other provinces in China will be required to assess whether our estimates of China-wide hospitalization costs of childhood depression based on Shandong are valid. Finally, when physicians rely on signs and symptoms listed in the ICD for depression, issues of inter-rater reliability may arise. Such inter-rater reliability issues may contribute to the absence of differences in depression between males and females.

## 5. Conclusions

We reported the hospitalization costs and estimated the financial burden on families of hospitalization expenses, for children with depression. Children with depression who had medical insurance experienced higher hospitalization costs and longer hospitalization stays than those without medical insurance. Uninsured patients experienced large OOP expenses, which plunged families into poverty, imposing on low-income families an especially severe financial burden. However, low income families—even with insurance—faced significant OOP expenses, which strained family budgets. The different hospital cost structures for drugs, treatment, bed fees, nursing and other costs, between insured and uninsured children with depression suggest the need for further investigations of treatment regimes. Specifically, parents of insured children may demand longer hospital stays and more intensive treatment for their children, including more drugs or tests, knowing that the bulk of such costs will be met through insurance. Additionally, doctors may over-service insured parents, providing more treatments, tests and drugs than necessary and, conversely, under-service uninsured patients without insurance to cover treatments or drugs. We recommend improved training for health professionals to identify depression in girls, and an awareness campaign about depression in girls to encourage girls and their families to seek treatment for depression. We recommend a further reform to China’s health insurance schemes as well, especially to allow migrant families to gain basic medical insurance.

## Figures and Tables

**Figure 1 ijerph-16-03526-f001:**
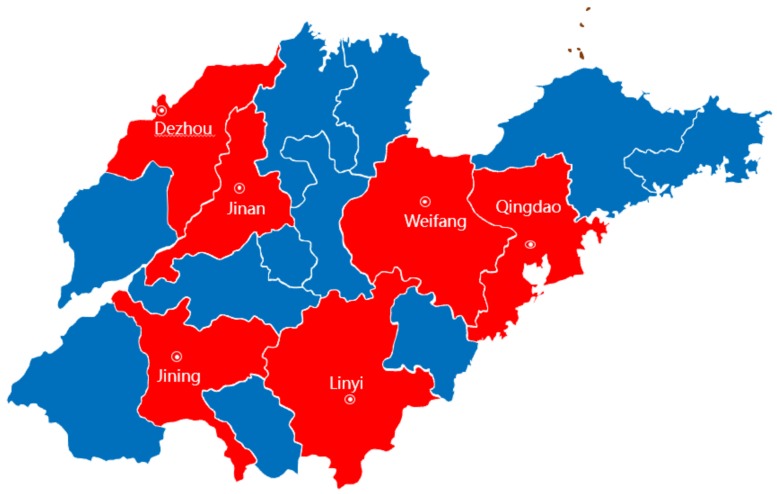
Sampling cities of this study in Shandong Province.

**Figure 2 ijerph-16-03526-f002:**
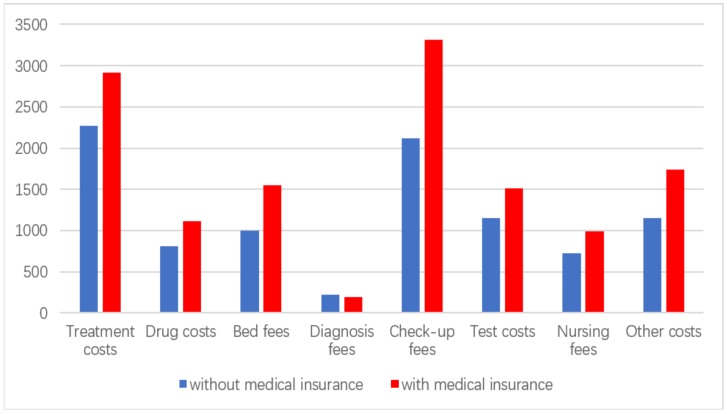
Hospitalization costs of children with depression with or without medical insurance (RMB).

**Table 1 ijerph-16-03526-t001:** Basic characteristic of children with depression.

Characteristics	Observation	%
**Age**		
9–11	145	29.71
12–14	151	30.94
15–17	192	39.35
**Sex**		
males	252	51.64
females	236	48.36
**Regional distribution**		
west	239	48.98
middle	173	35.45
east	76	15.57
**Comorbidities**		
yes	227	46.52
no	261	53.48
**Medical insurance**		
yes	364	74.59
no	124	25.41
**Type of health facility**		
public	438	89.75
private	50	10.25
**Total observations (%)**	**488**	100.00

**Table 2 ijerph-16-03526-t002:** Total costs, out-of-pocket and length of stay of children with depression.

	Total Costs (mean ± SD)	*p*-Value	Out-of-Pocket (mean ± SD)	*p*-Value	length of Stay (mean ± SD)	*p*-Value
**Sex**		0.5175		0.9206		0.2958
males	13844.46 ± 575.60		7637.34 ± 354.62		37.01 ± 1.86	
females	13312.61 ± 584.88		7585.10 ± 386.79		34.38 ± 1.67	
**Age**		0.5281		0.4206		0.4800
9-11	12905.82 ± 753.79		7086.04 ± 453.18		32.89 ± 2.21	
12-14	13689.74 ± 758.79		7890.88 ± 486.63		36.22 ± 2.20	
15-17	14021.27 ± 638.56		7790.07 ± 422.37		37.52 ± 2.10	
**Regional distribution**		0.4710		0.0905		**0.0011**
west	14095.90 ± 620.66		7644.54 ± 372.81		37.97 ± 1.84	
middle	13167.92 ± 682.49		8111.79 ± 460.93		37.36 ± 2.27	
east	12942.23 ± 844.70		6372.50 ± 571.83		25.03 ± 1.80	
**Comorbidities**		0.1389		0.0009		0.3367
yes	13021.02 ± 509.34		6802.17 ± 297.98		34.61 ± 1.65	
no	14238.29 ± 657.51		8543.30 ± 438.37		37.04 ± 1.92	
**Medical insurance**		**0.0001**		**0.0000**		**0.0000**
yes	14528.05 ± 490.26		6517.38 ± 239.81		38.87 ± 1.52	
no	10825.55 ± 674.39		10825.55 ± 674.39		26.54 ± 1.89	
**Type of health facility**		0.7908		**0.0065**		0.5582
public	13624.06 ± 428.55		7372.22 ± 263.69		36.00 ± 1.32	
private	13264.8 ± 1400.35		9713.26 ± 1049.14		33.56 ± 4.01	
Total	13587.25 ± 410.07		7612.08 ± 261.50		35.74 ± 1.26	

**Table 3 ijerph-16-03526-t003:** Regression coefficients and standard errors for major factors associated with hospitalization costs.

Factors	Unstandardized Coefficients	SE	Standardized Coefficients	*t*-Value	*p*-Value
Medical insurance	1911.308	187.191	0.184	3.923	**0.000**
Comorbidities	1087.368	829.169	0.060	1.311	0.190
Age	548.816	490.319	0.050	1.119	0.264
Type of health facility	1021.604	1390.286	0.034	0.735	0.463
Sex	395.350	811.625	0.022	0.487	0.626
Region	173.562	571.116	0.014	0.304	0.761

**Table 4 ijerph-16-03526-t004:** Hospitalization costs of children with depression (RMB).

Variable	All Patients (mean ± SD)	Patients with Insurance (mean ± SD)	Patients without Insurance (mean ± SD)	*t*-Value	*p*-Value
length of stay	35.74 ± 1.26	38.87 ± 1.52	26.54 ± 1.89	4.3525	0.0000
Total costs	13587.25 ± 410.07	14528.05 ± 490.26	10825.55 ± 674.39	3.9906	0.0001
Treatment costs	2748.97 ± 141.13	2913.32 ± 168.44	2266.54 ± 249.10	2.0012	0.0459
Drug costs	1036.58 ± 53.52	1112.57 ± 65.23	813.51 ± 85.00	2.4452	0.0148
Bed fees	1407.40 ± 55.55	1545.70 ± 66.73	1001.40 ± 87.80	4.3430	0.0000
Diagnosis fees	197.62 ± 21.44	188.91 ± 25.32	223.18 ± 42.74	0.6849	0.4937
Check-up fees	3011.26 ± 116.38	3314.98 ± 141.76	2119.70 ± 168.16	4.5615	0.0000
Test costs	1422.07 ± 48.08	1515.47 ± 58.68	1147.87 ± 73.27	3.3636	0.0008
Nursing fees	922.32 ± 44.23	989.24 ± 52.00	725.87 ± 81.45	2.6078	0.0094
Other costs	1591.54 ± 102.34	1740.25 ± 117.67	1155.00 ± 202.91	2.5031	0.0126
Out-of-pocket	7612.08 ± 261.50	6517.38 ± 239.81	10825.55 ± 674.39	7.5762	0.0000

**Table 5 ijerph-16-03526-t005:** China-wide hospitalization costs from childhood depression.

	Type	RMB	PercentageTotal Costs (%)
1	Total hospitalization costs—age <18, depression	6630578	0.016
2	Total hospitalization costs—all ages with depression	95236390	0.235
3	Total hospitalization costs—all diseases	40554194453

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
