# Peer review of "Hospitalization Costs and Financial Burden on Families with Children with Depression: A Cross-Section Study in Shandong Province, China"

_ijerph, 2019, doi:10.3390/ijerph16193526_

Round 1

Reviewer 1 Report

Hospitalization costs are a huge issue in health-care around the world and not much research has looked at this issue among depressed children and their families in China. As such, there is novelty to this manuscript and findings make an important contribution. Overall, this paper is interesting and well-written, however could be improved in a number of areas: 

1.) There are a number of grammar mistakes in this article. An example is on line 25 "...Children suffering depression" should be children suffering with or from depression. Please go back through and double check grammar errors throughout the manuscript. 

2.) Line 32 you state you used STATA 14, however elsewhere in the manuscript you site using STATA 15. Please correct. 

3.) Line 66 you state some researchers suggest disabilities in children are related to mental disorders. Can  you provide more context? What disabilities specifically? And are all mental disorders connected to these disabilities or are there some (i.e. depression) that are most closely linked? 

4.) Lines 79-91 point to research in this area providing support for the conduct of this study. But most of this data is from outside of China. These studies seem to have low relevance to what is happening in the China health system. Is there no data to report that comes from within the country? And if not it might be appropriate to state that. 

5.) In relation to the last point, the readers may benefit in the introduction from a little background on the China health system. You speak to this in the methods, but by then the reader is a bit confused. May help to introduce some of this in the background to help orient the reader to how the Chinese health system is set up. And how insurance rates have impacted other groups/illness categories in China if this info is available. 

6.) Line 239, "depression patients" is a bit troubling. We are trying to move away from stigmatizing language. Might suggests patients living with depression or with a depression diagnosis etc.

7.) Lastly, I think the discussion could use a bit more development. My initial thought when reading the findings was that is it makes sense to me the children with insurance have higher costs and longer hospital stays...because they have insurance that affords them the ability to have longer and more in depth services. Is it possible hospitals make choices to keep insured patients longer, and offer more costly services because they know insurance will cover the fee's? I am not certain this is accurate, but was my initial thought and I was surprised you did not provide this as a possible explanation beyond "over-serviced".

Author Response

Dear Editor

Thank you and the two reviewers for very helpful suggested revisions, which improve the paper. We have made all the revisions suggested using track changes in the text. Below we summary our response to each reviewer.

Response to Reviewer 1:

Hospitalization costs are a huge issue in health-care around the world and not much research has looked at this issue among depressed children and their families in China. As such, there is novelty to this manuscript and findings make an important contribution. Overall, this paper is interesting and well-written, however could be improved in a number of areas:

1.) There are a number of grammar mistakes in this article. An example is on line 25 "...Children suffering depression" should be children suffering with or from depression. Please go back through and double check grammar errors throughout the manuscript.

The paper has been re-read and corrected the paper for grammatical expression.

2.) Line 32 you state you used STATA 14, however elsewhere in the manuscript you site using STATA 15. Please correct.

In this study, the software used is STATA Version 14.0. This error has been corrected for consistency on line 161.

3.) Line 66 you state some researchers suggest disabilities in children are related to mental disorders. Can you provide more context? What disabilities specifically? And are all mental disorders connected to these disabilities or are there some (i.e. depression) that are most closely linked?

We have revised on line 70 as follows:

ORIGINAL

Some researchers argue that disabilities in children are related to mental disorders [12] and mental disorders underlie a substantial proportion of suicides, a leading cause of death in youths.[13]

REVISED

An Australian study found that mental disorders were the “largest” contributor to disabilities in adolescents, followed by chronic respiratory diseases and neurological disorders [12] and mental disorders underlie a substantial proportion of suicides, a leading cause of death in youths.[13]

4.) Lines 79-91 point to research in this area providing support for the conduct of this study. But most of this data is from outside of China. These studies seem to have low relevance to what is happening in the China health system. Is there no data to report that comes from within the country? And if not it might be appropriate to state that.

We searched the literature online, but no further Chinese studies were found. Lines 70-72 and 105-111 address the Chinese literature.

We have revised lines 108-111:

ORIGINAL

A Chinese study found that about 6.9% of total personal medical expenditure was attributable to depression and about 7.8% of total medical expenditure was attributable to the depressive symptoms,[23] which is almost three times as large as the impact of obesity on health care costs.[32]

REVISED

In one of the few Chinese studies, about 6.9% of total personal medical expenditure was attributable to depression and about 7.8% of total medical expenditure was attributable to the depressive symptoms,[23] which is almost three times as large as the impact of obesity on health care costs.[34]

5.) In relation to the last point, the readers may benefit in the introduction from a little background on the China health system. You speak to this in the methods, but by then the reader is a bit confused. May help to introduce some of this in the background to help orient the reader to how the Chinese health system is set up. And how insurance rates have impacted other groups/illness categories in China if this info is available.

Line 86-91 provides the following information on the hospital system in China:

There is a three-tiered public-private hospital system in China, comprising village clinics, township hospitals and county hospitals in rural areas, and community health centers, district hospitals and municipal hospitals in urban areas [24]. Village clinics, township hospitals and community health centers, provide preventive and primary care services; 100-499 bed county and district hospitals provide secondary specialty care and inpatient services; and large-scale city-based tertiary hospitals, with over 500 beds, provide complex healthcare. [25]

6.) Line 239, "depression patients" is a bit troubling. We are trying to move away from stigmatizing language. Might suggests patients living with depression or with a depression diagnosis etc.

We have made this change and also read the manuscript to correct any “stigmatizing language”.

7.) Lastly, I think the discussion could use a bit more development. My initial thought when reading the findings was that is it makes sense to me the children with insurance have higher costs and longer hospital stays...because they have insurance that affords them the ability to have longer and more in depth services. Is it possible hospitals make choices to keep insured patients longer, and offer more costly services because they know insurance will cover the fee's? I am not certain this is accurate, but was my initial thought and I was surprised you did not provide this as a possible explanation beyond "over-serviced"

We have addressed this point on lines 323-327 as follows:

ORIGINAL

The different hospital cost structures for drugs, treatment, bed fees, nursing and other costs, between insured and uninsured children with depression suggest the need for further investigations of treatment regimes, including over-demand by parents for treatment of their children, over-supply of treatment by medical staff and under-treatment of uninsured patients.

REVISED

The different hospital cost structures for drugs, treatment, bed fees, nursing and other costs, between insured and uninsured children with depression suggest the need for further investigations of the treatment regime. Specifically, parents of insured children may demand longer hospital stays and more intensive treatment for their children, including more drugs or tests, knowing that the bulk of such costs will be met through insurance. Also, doctors may over-service insured parents, providing more treatments, tests and drugs than necessary and, conversely, under-service uninsured patients without insurance to cover treatments or drugs.

Reviewer2:

I am uploading the document with comments and corrections. Overall this is a well-written study with very important subject matter. I see one problem in the Method section in that you do not discuss the qualifications of the physicians. If you have that info, you could add it to the table. This info is relevant for the skill of diagnosing depression. Specifically, it appears that the physicians are relying on signs and symptoms listed in the ICD for depression. We know from previous research that there is low inter-rater reliability when relying solely on a diagnostic list. Furthermore, we know that physicians will vary in skill as diagnosticians based on their years of experience with the population. Later you mention the difference in this study from previous studies in finding no difference in rates of depression between males and females- again not knowing if there was rater bias among the physicians limits the strength of your conclusions. I also made a suggestion for the limitations and conclusions - considering other factors for the difference in older larger studies' rates of depression between men and women.

 Unfortunately, there are no data on the qualifications of physicians. We appreciate the reviewer drawing our attention to inter-rater reliability when relying on a diagnostic test. We have added to the Limitations at line 307-310 the following:

Finally, when physicians rely on signs and symptoms listed in the ICD for depression, issues of inter-rater reliability may arise. Such inter-rater reliability issues may contribute to the absence of differences in depression between males and females.

Jian Wang, Professor

Dong Fureng Institute of

Economic and Social

Development

Wuhan University

Reviewer 2 Report

I am uploading the document with comments and corrections. Overall this is a well-written study with very important subject matter. I see one problem in the Method section in that you do not discuss the qualifications of the physicians. If you have that info, you could add it to the table. This info is relevant for the skill of diagnosing depression. Specifically, it appears that the physicians are relying on signs and symptoms listed in the ICD for depression. We know from previous research that there is low inter-rater reliability when relying solely on a diagnostic list. Furthermore, we know that physicians will vary in skill as diagnosticians based on their years of experience with the population. Later you mention the difference in this study from previous studies in finding no difference in rates of depression between males and females- again not knowing if there was rater bias among the physicians limits the strength of your conclusions. I also made a suggestion for the limitations and conclusions - considering other factors for the difference in older larger studies' rates of depression between men and women.

Thank you for conducting this important study. Hope my comments help

Author Response

 Unfortunately, there are no data on the qualifications of physicians. We appreciate the reviewer drawing our attention to inter-rater reliability when relying on a diagnostic test. We have added to the Limitations at line 307-310 the following:

Finally, when physicians rely on signs and symptoms listed in the ICD for depression, issues of inter-rater reliability may arise. Such inter-rater reliability issues may contribute to the absence of differences in depression between males and females.